# The Brainstem Cavernoma Case Series: A Formula for Surgery and Surgical Technique

**DOI:** 10.3390/medicina59091601

**Published:** 2023-09-05

**Authors:** Marcos Tatagiba, Guilherme Lepski, Marcel Kullmann, Boris Krischek, Soeren Danz, Antje Bornemann, Jan Klein, Antje Fahrig, Tomaz Velnar, Guenther C. Feigl

**Affiliations:** 1Department of Neurosurgery, University of Tuebingen Medical Center, 72074 Tübingen, Germany; 2Department of Neuroradiology, University of Tuebingen Medical Center, 72074 Tübingen, Germany; 3Department of Neuropathology, University of Tuebingen Medical Center, 72074 Tübingen, Germany; 4Institute for Medical Image Computing, Fraunhofer MEVIS, 28359 Bremen, Germany; 5Department of Radiotherapy and Radiooncology, General Hospital Klinikum Bamberg, 96049 Bamberg, Germany; 6Department of Neurosurgery, University Medical Centre Ljubljana, 1000 Ljubljana, Slovenia; 7Department of Neurosurgery, General Hospital Klinikum Bamberg, 96049 Bamberg, Germany

**Keywords:** cavernous malformation, brainstem, surgery, electrophysiology, fibre tracking

## Abstract

*Background and Objectives*: Cavernous malformations (CM) are vascular malformations with low blood flow. The removal of brainstem CMs (BS) is associated with high surgical morbidity, and there is no general consensus on when to treat deep-seated BS CMs. The aim of this study is to compare the surgical outcomes of a series of deep-seated BS CMs with the surgical outcomes of a series of superficially located BS CMs operated on at the Department of Neurosurgery, College of Tuebingen, Germany. *Materials and Methods*: A retrospective evaluation was performed using patient charts, surgical video recordings, and outpatient examinations. Factors were identified in which surgical intervention was performed in cases of BS CMs. Preoperative radiological examinations included MRI and diffusion tensor imaging (DTI). For deep-seated BS CMs, a voxel-based 3D neuronavigation system and electrophysiological mapping of the brainstem surface were used. *Results:* A total of 34 consecutive patients with primary superficial (*n* = 20/58.8%) and deep-seated (*n* = 14/41.2%) brainstem cavernomas (BS CM) were enrolled in this comparative study. Complete removal was achieved in 31 patients (91.2%). Deep-seated BS CMs: The mean diameter was 14.7 mm (range: 8.3 to 27.7 mm). All but one of these lesions were completely removed. The median follow-up time was 5.8 years. Two patients (5.9%) developed new neurologic deficits after surgery. Superficial BS CMs: The median diameter was 14.9 mm (range: 7.2 to 27.3 mm). All but two of the superficial BS CMs could be completely removed. New permanent neurologic deficits were observed in two patients (5.9%) after surgery. The median follow-up time in this group was 3.6 years. *Conclusions*: The treatment of BS CMs remains complex. However, the results of this study demonstrate that with less invasive posterior fossa approaches, brainstem mapping, and neuronavigation combined with the use of a blunt “spinal cord” dissection technique, deep-seated BS CMs can be completely removed in selected cases, with good functional outcomes comparable to those of superficial BS CM.

## 1. Introduction

The Cavernous malformations (CM) are angiographically hidden vascular lesions with low flow [1]. They account for 10% to 15% of all vascular malformations [2]. Approximately 20% of all CM of the central nervous system are located in the brainstem [3,4,5]. Although cavernous malformations of the brainstem (BS CM) have a low incidence and are benign lesions, studies of their natural history have shown that they may be associated with a high risk of hemorrhage, causing potentially devastating neurologic deficits. In the past, BS CMs were considered inoperable because of the high surgical morbidity associated with removal of these lesions. Microsurgical techniques have improved considerably since the first removal of a BS CM by Dandy in 1928, and in the last two decades several authors have published promising surgical results in patients with BS CMs [3,5,6,7,8,9,10,11,12,13]. Nevertheless, the treatment of these lesions remains a major challenge, especially when the lesion is located deep in the brainstem. Given the available data on the natural history of BS CMs, there is an ongoing debate about the timing and method of treatment. Data on conservative strategies with observation, radiosurgery, and microsurgical excision can be found in the literature [3,4,10,11,12,14,15,16,17,18,19,20,21,22,23,24,25,26,27,28]. Considering the natural history of BS CMs, with a risk of bleeding ranging from 0.6% to 60% in some series, depending on the previous occurrence of a bleeding episode and surgically induced morbidities, it becomes clear how difficult it is to choose the right timing and appropriate treatment for each individual patient, especially in cases with deep-seated BS CMs [4,8,10,11,14,15,22,29].

In this study, the surgical outcome of patients with deep-seated BS CMs is analysed and compared to the outcome of a series of superficial brainstem cavernomas. The authors sought to define the factors that influenced the decision-making process for surgical treatment and for the timing of surgery in patients with deep-seated and superficial BS CMs.

## 2. Materials and Methods

The study was designed and conducted in the Department of Neurosurgery, College of Tuebingen, Germany. In this case series study, we collected data from a number of patients with deep-seated BS CMs and their surgical outcomes after surgery. The operations were performed over a period of six years. Medical records and radiological images were reviewed. Surgical approaches, complications, and treatment outcomes were analysed. All patients were fully informed and gave written informed consent after approval from the Ethical Institutional Review Board (Ethical Board number 3 May 2022).

Thirty-four consecutive patients with BS CMs who had undergone surgery by the lead neurosurgeon (M.T.) were included in this retrospective study. Informed consent was obtained from all patients. Medical records, operative reports, and the results of neurologic examinations during the follow-up period were used to determine and compare the functional outcomes of surgical treatment in patients with deep-seated and in patients with superficial BS CMs. Based on these results and the reviewed decision process for surgical treatment, a flowchart (Figure 1) was developed, summarising the decision process for surgical and conservative treatment in these patients.

Complete surgical excision of the lesion was determined based on neurosurgeon’s assessment and the absence of residuals on postoperative MR imaging within 24 h and then 3 months after surgery. Neurologic examinations were performed before and within 1 week after surgery to assess existing deficits and accentuated or new neurologic deficits after surgery. Follow-up examinations of all 34 patients were performed according to the management protocols of the Department of Neurosurgery in Bamberg on the first postoperative day, before discharge, 3 months after surgery, and during follow-up examinations thereafter.

### 2.1. Imaging

Preoperative and postoperative imaging included T1 3D magnetization-prepared rapid gradient echo (MP-RAGE) MRI with and without gadolinium contrast enhancement for neuronavigation and T2 and T2* sequences to visualize the BS CM and signs of haemorrhage. Cavernomas more than 2 mm from the surface of the brainstem were defined as deep-seated BS CMs. Diffusion tensor imaging (DTI) was performed in all cases operated on after January 2008 (*n* = 19) to visualize the fibrous tracts of the brainstem surrounding the BS CM. Imaging data were analysed offline for fibre tracking and then loaded into a neuronavigation system for intraoperative use. Follow-up examinations were performed within 24 h after surgery and then 3 months after surgery to verify complete removal. MeDigS^®^-Archiv V1.1 (FMS; Graz, Austria) was used to manage patient data and digital image data.

### 2.2. Intraoperative Electrophysiological Recordings

All operations were planned preoperatively and then performed using neuronavigation (CBYON™—Med-Surgical Services Inc., Sunnyvale, CA, USA) to select the shortest path to the deep-seated lesions that were not visible at the brainstem surface. The CBYON™ Neuronavigation System provides three-dimensional (3D) volumetric image rendering and allowed for the modulation of the opacity of tissue layers, making the brainstem transparent on the navigation screen (Figure 2). Intraoperative electrophysiological mapping of the brainstem surface was performed during all procedures to find the safest entry point. Continuous intraoperative monitoring (IOM) during the procedure included the recordings of motor-evoked potentials (MEP) and sensory-evoked potentials (SEP) of the lower and upper limbs, and the recordings of auditory-evoked potentials of the brainstem (BAEP) and the MEP of the facial nerve in all cases. Depending on the level of the lesion in the brainstem, electromyography (EMG) and MEP recordings of the following cranial nerves (CN) were also performed: CN III for midbrain lesions, CN VI for pontine lesions, and CN IX–XII for medulla oblongata lesions. MEPs were recorded continuously with transcranial electrical stimulation via corkscrew electrodes positioned at CZ and C3 +1 cm lateral or C4 +1 cm lateral for the facial nerve and C3 and C4 for the upper and lower extremities. Stimulations were always contralateral to the affected side for 0.5 milliseconds, with five pulses and a voltage range of 130–300 V for cranial nerves (CN) or 300–500 V for upper and lower extremities, always with 2 milliseconds between stimuli. MEPs were recorded from needles placed in the affected target muscles. Baseline values for SEP and MEP were obtained before craniotomy. Direct stimulation of the brainstem was performed using a bipolar probe with 0.3 to a maximum of 3 mA, 3.2 to 5.2 Hz continuous stimulation, and a stimulation duration of 0.1 milliseconds.

### 2.3. Surgical Approaches

Nearly all brainstem lesions were removed via infratentorial approaches to the posterior fossa. Table 1 summarizes all approaches according to brainstem lesions. Subtemporal approaches were not used in any case in this series. Two basic posterior fossa craniotomies were used: the retrosigmoid (lateral suboccipital) craniotomy and the medial suboccipital craniotomy. The retrosigmoidal lateral approach (Figure 3) was used to access the CPA, the (antero)lateral pons, and the (antero)lateral medulla oblongata. The retrosigmoidal supracerebellar approach was used to reach the lateral midbrain. The medial suboccipital supracerebellar approach was used to reach the dorsal midbrain and quadrigeminal plate. The medial suboccipital subtonsilar transventricular approach was used to reach the dorsal pons and dorsal medulla oblongata through the fourth ventricle. The medial suboccipital approach was used to reach lateral lesions of the medulla oblongata and the foramen Luschka.

### 2.4. Access to the Brainstem: The Spinal Cord Dissection Technique

To enter the brainstem, the specific microsurgical technique involved a sharp incision of the pial brainstem surface and the blunt dissection and separation of the deep brainstem tissue with microdissectors (Figure 4A). This is very similar to the surgical technique used to treat intramedullary lesions. Therefore, we call this surgical technique the “spinal cord dissection technique”. We have used the same technique in the successful removal of a cavernous haemangioma from an optic nerve with the preservation and improvement of vision [30]. In this technique, the dissection of the lesion is performed bimanually with the continuous irrigation of the surgical field with Ringer’s solution. The viscoelasticity of the brainstem tissue facilitates blunt dissection in the direction of the fibres, allowing for the gentle widening of the opening in the brainstem and safe access to the cavernoma (Figure 4B). Once the cavity of the lesion is opened, old fluid blood is drained from the cavernoma, creating sufficient space for the microsurgical resection of the cavernoma. The resection is then continued bimanually using micro-tumour forceps to hold the thin hyalinized capsule of the BS CM and micro-bayonet dissection forceps to gradually separate the lesion from the brainstem tissue (Figure 4C). This is performed under constant irrigation, which allows for a clear view of the surgical field during dissection. The plane of dissection lies between the hyalinized capsule of the BS CM and the inner layer of the brainstem, which is typically stained with hemosiderin. BS CMs are often associated with venous abnormalities that must remain intact during removal of the cavernoma.

## 3. Results

A total of 14 men (41.2%) and 20 women (58.8%) with a mean age of 42 years (range: 3.7 to 77.4 years) underwent surgery for BS CMs. All patients had at least one bleeding episode and suffered from a neurological deficit at the time of surgery. Multiple bleeding episodes before surgery occurred in 11 patients (32.4%), and two of these patients suffered from deep-seated BS CM. All the removed lesions were sent for pathohistological assessment, which is standard procedure when removing pathological tissue during brain surgery. In all cases, it was confirmed that the excised lesions were cavernomas.

The symptoms and clinical signs on admission and after surgery for both groups of patients are summarized in Table 2. The location of the resected lesions included the midbrain (*n* = 10; 29.4%), pons (*n* = 18; 52.9%), and medulla oblongata (*n* = 6; 17.6%). Complete tumour removal of BS CMs (deep-seated and superficial) was achieved in 31 patients (91.1%). The following surgical approaches were used: lateral supracerebellar transtentorial approach (*n* = 8; 23.5%), medial supracerebellar infratentorial approach (*n* = 3; 8.8%), retrosigmoid lateral approach to the CPA (*n* = 16; 47.1%), and medial suboccipital approach to the fourth ventricle (*n* = 7; 20.6%). Two illustrative cases showing images of patients with a deep-seated and a superficial BS CM are shown in Figure 5, Figure 6, Figure 7 and Figure 8. Six patients (17.6%) suffered from multiple cavernous malformations of the central nervous system. The median interval between the first neurological symptoms due to haemorrhage and surgery was one month.

Patients who were under observation and did not undergo surgery were not included in this series because surgical outcome was the focus of this study. However, during the same period, nine patients (seven men/two women) with small, deep-seated BSCs were observed clinically and radiologically.

### 3.1. Deep-Seated BS CMs (n = 14)

One woman with a deep-seated cavernoma in the medulla oblongata was pregnant at the time of surgery. One patient had undergone two previous surgeries with the incomplete removal of the BS CM before being referred to our institution. The median of the shortest distance between the lesion and the surface of the brainstem was 2.9 mm in deep-seated BS CMs. The median diameter in this group (measured on axial T1-weighted MRI scans with contrast enhancement) was 14.7 mm (range, 8.3 to 27.7 mm). All but one of these lesions were completely removed. The median follow-up time was 5.8 years. The Modified Ranking Scale (MRS) for patients with deep-seated BS CMs was 1 before and 2 after surgery (Table 3) [31]. In four patients (11.7%), preoperative neurologic symptoms worsened in the immediate postoperative period, and in two of these patients (5.9%), symptoms did not regress during the follow-up period. One of these patients developed degenerative motor fibre disease postoperatively with progressive neurologic deterioration due to motor fibre degeneration during the follow-up period. However, the reasons for this degenerative disease remained unclear.

### 3.2. The Tuebingen Brainstem Cavernoma Equation

Decision making on when to operate on deep-seated BS CMs was defined using the “Tuebingen brainstem cavernoma equation”. The largest diameters of BS CMs and the distance to the surface were measured in all patients. We summarized our results in a mathematical equation (Figure 9). We found that when the distance of the lesion to the surface (*D*) of the brainstem was equal to or less than half of the largest diameter (*ld*) of the lesion measured on an axial T1 scan, surgical access and safe removal of the deep-seated BS CMs was possible.

### 3.3. Superficial BS CMs (n = 20)

The mean diameter (measured on axial T1-weighted MRI scans with contrast enhancement) was 14.9 mm (range, 7.2 to 27.3 mm). Only two large superficial BS CMs with extensive calcifications could not be completely removed. In the immediate postoperative period, six patients (17.6%) experienced a transient worsening of preoperative neurologic deficits. New permanent neurological deficits after surgery were observed in two patients (5.9%). The MRS for patients with superficial BS was 2 before and 3 after surgery. The median follow-up time in this group was 3.6 years.

### 3.4. Neurological Deficits after the Operation

To assess the neurological symptoms in both groups of cavernoma patients, the deep-seated and the superficial, the Modified Ranking Scale (MRS) was used. The rate of neurological symptoms was assessed preoperatively and during follow-up (Table 3).

## 4. Discussion

### 4.1. Indication and Timing for Surgical Treatment

BS CMs have been shown to have a particularly poor natural history with a significantly higher rate of severe neurologic deficits and a higher rate of recurrent symptomatic haemorrhage compared to supratentorial CM [10,15,22,32]. Therefore, the goal of surgery should be to stabilize the situation and prevent rebleeding to avoid disease progression. The risk of rebleeding in patients with a BS CM has been reported in some series to be 0.6% to 60% per year, depending on previous bleeding [4,7,8,10,11,15,22,29,33,34].

BS CMs are a unique subgroup of cavernous malformations. Although the statistical possibility for CM haemorrhage varies according to the literature, it is typical for the brainstem location to have higher haemorrhagic presentations, as well as technical challenges for their removal [35,36,37]. Statistically, the average annual rate of CM haemorrhage is 0.7% to 1.1% per lesion in patients with no previous haemorrhage. This risk increases (to almost 4.5% and up to 60%) in patients who have had a previous lesion bleed [10,11,15,22,35]. Approximately three to four years after a haemorrhagic event, the risk of further bleeding decreases. Of course, the risk of bleeding also depends on other factors, such as gender, the presence of a developmental venous anomaly, and the location of the CM [35,38]. As mentioned earlier, CMs located in the brainstem are more likely to bleed [36]. According to the study by Kong et al. [38], the 5-year cumulative risk of haemorrhage was 30.8%, was higher in subgroups when risk factors were present that helped predict potential bleeding in patients, and were useful for treatment decision-making. According to known statistics, we recommend CM surgical removal in patients with risk factors that increase the possibility of bleeding. These include previous symptomatic bleeding, a positive family history, gender, and age [37,39]. All our operated patients had suffered at least one acute or previous bleeding episode. Therefore, according to the known literature, the risk of rebleeding in this group remains high and surgical treatment was recommended. This outweighed the potential risk of postoperative neurological impairment that may occur during surgery. In addition, all patients already had some neurological deficit due to the haemorrhage itself. Since the surgical mortality is reported to be 1.9%, the recurrence rate is more than 50% to 60%, which outweighs the risk even though surgery is performed in a problematic area such as the brainstem [10,23].

### 4.2. Surgical Treatment of Deep-Seated and Superficial BS CMs

In patients with deep-seated BS CMs or with an incidental finding of such lesion, recommendations vary widely in the literature. In these patients, we recommend surgery only if the risk of surgical resection causing neurologic deficits is statistically lower than the risk of deterioration due to haemorrhage based on the natural history of these lesions. This means that in asymptomatic patients with deep-seated BS CMs, we recommend surgery only if there is evidence of previous haemorrhage on a T2*-weighted MRI sequence and if the lesion is located near the pial or ependymal surface, where it can be reached via a dorsal approach while avoiding motor fibres [3].

The results of this study demonstrate that the surgical and functional outcomes of patients with deep-seated BS CMs and those with superficial BS CMs may be comparable when the appropriate surgical strategy is used. The “spinal cord dissection technique” described has also been shown to be very helpful in removing lesions in much smaller structures than the brainstem, such as the optic nerve. In this case, the identical technique was successfully used by the lead neurosurgeon in this study (M.T.) to remove a cavernous haemangioma, which not only preserved the optic nerve, but also improved the patient’s vision [30]. The comparable results between the two groups of patients are also evidenced by the improved median total score in both groups. The MRS was 1 before and after surgical treatment in patients with deep-seated BS CMs and 2 before and 3 after surgical treatment in patients with superficial BS CMs. The rate of new neurological symptoms was also not significantly different between these two groups, as shown in Table 3. In addition, complete removal could not be achieved in only one patient (2.9%) with a deep-seated BS CM compared to two patients (5.8%) with superficial BS CMs. This surgical outcome compares well with the results of other groups reporting the incomplete removal of deep-seated BS CMs in 7.7% of surgically treated cases, and justifies the recommendation of surgical treatment of deep-seated BS CMs because the risk of rebleeding is much higher [7]. In a comprehensive review of 46 surgical series analysing a total of 745 patients with BS CMs, complete removal was reported in 92%, although it was not described how many of these patients had deep-seated BS CMs [23]. This is also consistent with our results, in which complete tumour removal was achieved in 91.2% of our patients. In addition, there was no mortality in our series compared to the 1.9% surgical mortality reported by Gross et al. in their literature review and other series, which can be considered positive for our results [10,23,40,41].

An analysis of our patients with deep-seated and superficial BS CMs has led us to establish the “Tuebingen brainstem cavernoma equation”, according to which the shortest distance (*D*) of the BS CM to the surface of the brainstem should be equal to or less than 1/2 the largest diameter of the lesion (*ld*), measured on an axial T1 scan, to ensure safe microsurgical removal [*D* ≤ *ld*/2] using the described spinal cord blunt dissection technique. We have found that this equation best describes whether or not a deep-seated BS CM can be removed with very little surgical morbidity.

We must emphasise that the proposed formula is not absolute and represents an initial starting point for further evaluations in order to find a more objective, systematic and logical mathematical way to make a decision for surgery. The formula is meant to be the initial step for easier decisions making regarding the surgical indications of BCs. As we mention in our case series, further studies need to be performed in order to evaluate this formula.

Although Wang et al. have shown that lesion size does not correlate with surgical outcome, it is evident that the resection of small lesions is considerably more difficult, especially in deep-seated BS CMs [13]. The median size of deep-seated BS CMs was 14.7 mm (range: 8.3–27.7 mm) and 14.9 mm (range: 7.2–27.3 mm) for superficial BS CMs. The median of the shortest distance between the lesion and the surface of the brainstem was 1.5 mm for superficial and 2.9 mm for deep-seated BS CMs. Surgical morbidity in this series was 11.7% during long-term follow-up, with new permanent neurologic deficits observed in two patients (5.9%) with deep-seated and in two patients (5.9%) with superficial BS CMs. This indicates that there was no difference in functional outcome between these two groups of patients in our series. Therefore, the overall functional outcome of both groups in our series is consistent with the results published by other authors, in which surgical morbidity ranges from 12% to 27.7%, but not only for deep-seated BS CMs [3,8,13,23].

### 4.3. Electrophysiology, Neuronavigation, and Ultrasonography

After the exposure of the brainstem, systematic mapping of its surface was performed to determine the safest entry point. Several authors have described in detail different methods of brainstem mapping [28,42,43,44]. We recommend performing brainstem mapping even in the presence of discoloration of the brainstem surface. In addition to standard brainstem mapping and monitoring, we performed transcranial electrical stimulation in all cases to monitor motor evoked potentials (MEP) of muscle groups of the arms, legs, and relevant CNS in each individual procedure. This is highly recommended, especially during the approach to the lesion, to ensure that no deficits are caused during the approach to the lesion. A voxel-based neuronavigation system with 3D image volume display and a feature that allows for a variation in tissue layer opacity (CBYON™—Med-Surgical Services Inc., Sunnyvale, CA, USA) was used as needed. An “endoscopic view” allowed for the direct localization of the lesion on BS, especially in cases where the lesion did not reach the surface (Figure 2).

### 4.4. Intraoperative Strategy

Access via the posterior fossa provided excellent exposure of the posterior and even the anterolateral portion of the brainstem, pons, and medulla oblongata [3,8,10,12,45]. Brown et al. described a two-point method for planning the best surgical approach to brainstem lesions, in which the shortest distance between the centre and the surface of the brainstem determines the direction of surgical approach [46]. However, this formula does not always indicate the safest point of access to the brainstem lesion because approaching a BS CM anteriorly would require microsurgical dissection through motor fibres, which greatly increases the risk of causing neurologic symptoms just by approaching the lesion. We used the lateral supracerebellar transtentorial (*n* = 8/23.5%), medial supracerebellar infratentorial (*n* = 3/8.8%), retrosigmoid approach to the CPA (*n* = 16/47.1%), and medial suboccipital subtonsilar approach to the fourth ventricle (*n* = 7/20.6%), (ventricle (*n* = 7/20%)). The advantages of the above approaches, especially when performed in the semi-sitting position, are obvious. The semi-sitting position allows for better anatomic orientation, as the view of the brainstem is straight and fluids and blood do not obscure the field of view. In addition, no ventricular drainage is required because the opening of the basal cistern provides sufficient relaxation of the brain, which allows for the sufficient retraction of the cerebellum to expose the brainstem. In addition, the compression of the temporal lobe and injury to the Labbé vein can be completely avoided.

### 4.5. Stereotactic Radiosurgery as an Alternative Treatment Option

Stereotactic radiosurgery (SRS) has been used as an alternative treatment modality for cerebral cavernous malformations, but the number of patients treated is much smaller than that of surgical series, and the results are not comparable to those of surgical series [16,17,18,19,20,29]. Nevertheless, some authors still emphasize the advantages of SRS compared to surgical treatment. However, recent data show that surgical treatment, especially by experienced neurosurgeons, results in much lower morbidity than SRS in these lesions [17,18,20]. SRS relies on accurate visualization of the lesion, which is particularly difficult when haemorrhage has occurred because the lesion appears larger on MR images due to surrounding blood [13]. In these cases, the exact contour of the cavernoma may be obscured by the hemosiderin shadow of the blood, making accurate irradiation impossible. This may be the reason why the postoperative bleeding rate after gamma knife radiosurgery ranges from 1.6% to 15% [20,29]. After a detailed review of the available studies on SRS, the percentage of complications after radiation was recently reported by Steiner et al. to be 19%, with half of the neurological deficits being permanent [47]. This is significantly higher than the rebleeding rate after complete microsurgical excision, which is 0%. In addition, deaths have also been reported in a series of patients treated with SRS, indicating that this risk remains. It is fair to say that based on the available data, SRS cannot currently be considered a better treatment alternative to well-performed microsurgical resection. This conclusion is also reached by experts in the field of SRS such as Lindquist et al., who stated that the SRS treatment of BS CMs is not efficient enough to be recommended as a treatment modality for these benign lesions [19]. This finding was also confirmed by Steiner et al. who have extensive experience with SRS [47].

### 4.6. Possible Complications after the BS CMs Surgery

The brainstem is a difficult anatomical region and neurological sequelae are always possible due to both the lesion itself and the surgical procedure. The most common surgical complications are the standard complications that can occur with almost any neurosurgical procedure. In general, surgical complication rates vary between 2% and 9% [48]. These include a mortality rate of 1.5%, a wound infection rate of 1.5% and a bleeding rate of 4.5% at the surgical site. Postoperative medical complications can occur in 3% to 9% of patients [48,49]. Complications related to the cranial nerves, sensory and motor pathways, cerebellar mutism, and cerebrospinal fluid circulation are possible with brainstem surgery [49]. In our patients, the complications we noted resulted in headache, hypaesthesia, gait disturbances, cranial nerve deficits, the incomplete removal of the CM, hydrocephalus, and degenerative motor fibre disease of unknown cause.

An important and rarely reported complication is olive degeneration [50,51]. It is a relatively common consequence of posterior fossa surgery, especially in children treated for high-grade tumours [50]. Olive degeneration is caused by lesions of the posterior fossa or brainstem involving the dentato–rubro–olivaria pathway, also known as the Guillain–Mollaret triangle. Clinical and radiological features of this disorder include ocular myoclonus, gaument tremors, nystagmus, ataxia, and vertigo. MRI shows the T2 hyperintensity of the inferior olive complex on magnetic resonance imaging. The symptomatic degeneration of the olives can complicate recovery in patients with posterior fossa or brainstem lesions. It most commonly occurs after posterior fossa surgery, although other cases are also important, such as posterior fossa tumours, traumatic brain injury, and cerebellar infarction. The most common reported cause is posterior fossa surgery secondary to cavernomas. Although most patients may remain asymptomatic, the degeneration of the orbit complicates postoperative recovery [50,51,52]. In our patient group, we did not directly evaluate olive degeneration clinically. However, complications in our group included gait disturbance and diplopia, so some of the cases might be related to some kind of olive degeneration. Because disruption of the dentato–rubro–olivary tract can complicate patient recovery, neurosurgeons should be aware of this complication to avoid misdiagnosis and unnecessary investigations.

### 4.7. The Limitations of Our Study

Of course, the study also has some limitations. The first limitation is the relatively small number of patients. However, we can say that CMs in the brainstem are not a common pathology compared to CMs in supratentorial areas and cerebellum. The patients with BS CMs represent a specialised subgroup that tends to settle in specialised and dedicated neurosurgical centres that frequently treat such pathologies, as was the case in this study. The solution here would be to expand the retrospective assessment framework and thus capture more patients. Secondly, although our patients were diagnostically well prepared for surgery, DTI, which is of great importance in brainstem CMs, was performed only in patients operated on after January 2008. This technique was unfortunately not available in our clinical department before. We believe that safety would increase if diagnostic imaging were even better. Third, as mentioned above, the Tuebingen brainstem cavernoma equation helps in deciding when deep-seated BS CMs should be operated on. However, it must be interpreted and applied with caution and is not a simple formula for choosing surgery. Each patient must be interpreted individually, taking into account all risk factors, including family history, the risk of rebleeding, the location of the CM, and the consideration of possible neurological sequelae that may result from surgery. Fourth, we can focus more on the new complications following surgery in such challenging anatomical areas, such as olive degeneration, as described earlier.

## 5. Conclusions

The results of this study show that by using the “spinal cord dissection technique” in combination with minimally invasive posterior fossa approaches, brainstem mapping, and neuronavigation in the hands of an experienced team, both superficial and deep-seated BS CMs can be completely removed in the majority of cases with a good functional outcome. The current data do not indicate that there are alternative treatments for these brainstem lesions that provide similarly good results.

## Figures and Tables

**Figure 1 medicina-59-01601-f001:**
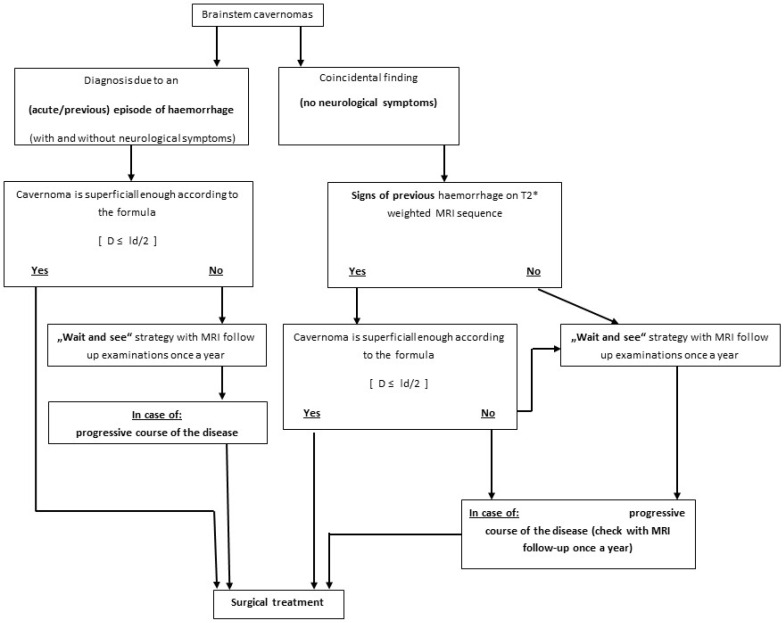
Flowchart describing the decision-making process for surgical and conservative treatment of patients with BS CM, including the “Tuebingen brainstem cavernoma equation”, where (*D*) is the distance from the lesion to the surface of the brainstem, which must be equal to or less than half the largest diameter (*ld*) of the lesion when measured on an axial T1 image. For progressivity check (the right part of the diagram), an MRI is essential. It is not possible to reach a decision without MRI assessment. * is the designation of imaging.

**Figure 2 medicina-59-01601-f002:**
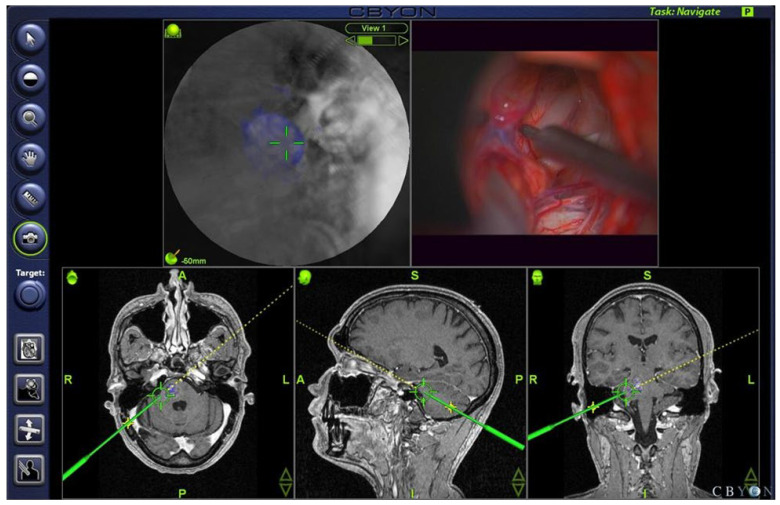
Screenshot of intraoperative neuronavigation (CBYON™—Med-Surgical Services Inc., Sunnyvale, CA, USA) showing the “virtual endoscopic” view (**top left** image) through the tip of the pointer (**top right** image) with 3D volume image rendering and modified opacity of the tissue layers, revealing the deep-seated cavernoma (blue) in the pons and guiding the neurosurgeon to a safe entry point.

**Figure 3 medicina-59-01601-f003:**
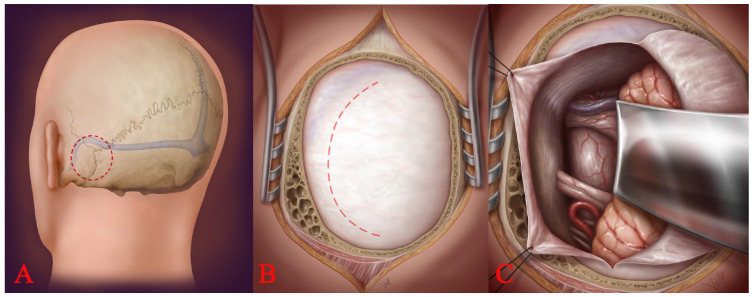
Location of the craniotomy (**A**), transection of the dura (**B**), and insertion of the spatula (**C**).

**Figure 4 medicina-59-01601-f004:**
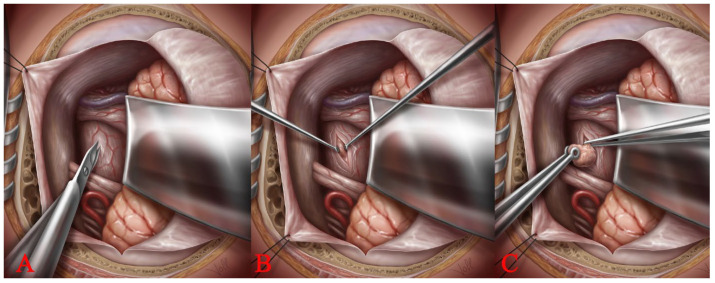
The following figures show the sequential steps of microsurgical resection of the cavernoma through a retrosigmoidal approach to the cerebello-pontine angle. The opening of the pial surface of the brainstem with microscissors (**A**). The bimanual opening and expansion of the brainstem with two microdissectors by blunt dissection (**B**). The viscoelasticity of the brainstem facilitates blunt dissection and allows for the gentle and safe access to the cavernoma. Micro-tumour forceps holding the cavernous malformation and micro-bayonet dissection forceps and carefully separating the cavernous malformation from the brainstem tissue (**C**).

**Figure 5 medicina-59-01601-f005:**
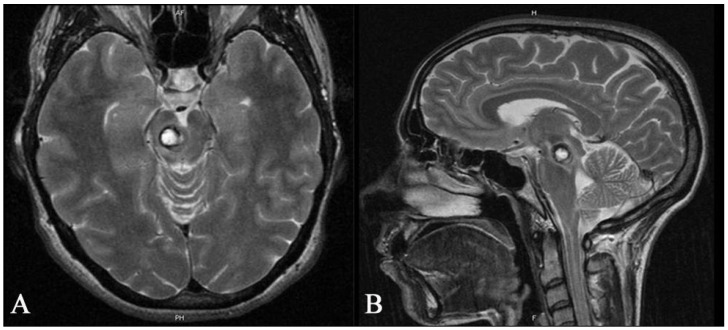
Preoperative axial (**A**) and sagittal (**B**) T2 MR images of a 35-year-old woman showing a deep-seated cavernous malformation in the midbrain on the right side. The patient had been suffering from headaches and gait disturbances for several weeks.

**Figure 6 medicina-59-01601-f006:**
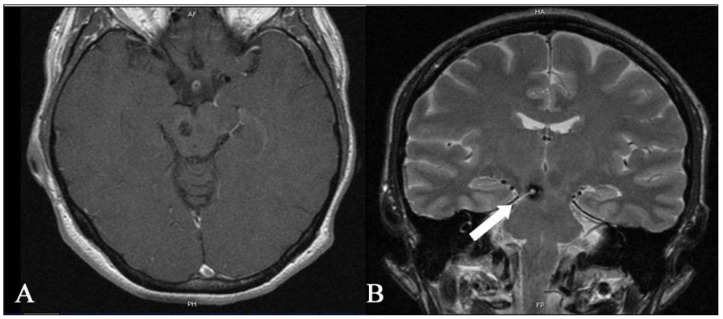
Postoperative axial (**A**) T1 and coronal T2 images (**B**) MR show that complete removal of the deep-seated cavernous malformation of the brainstem was achieved. The coronal image on the right shows the entry site into the brainstem (white arrow) through which the cavernoma was accessed. The patient was neurologically stable after surgery. She had no new neurologic deficits after surgery, either in the immediate postoperative period or during the 3-month follow-up.

**Figure 7 medicina-59-01601-f007:**
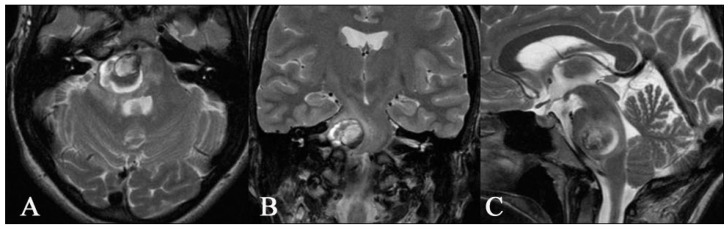
Preoperative axial (**A**), coronal (**B**), and sagittal (**C**) T2 images MR of a 35-year-old woman showing a superficial cavernoma in the pons on the right side. The patient suffered from headaches and gait disturbances for several weeks.

**Figure 8 medicina-59-01601-f008:**
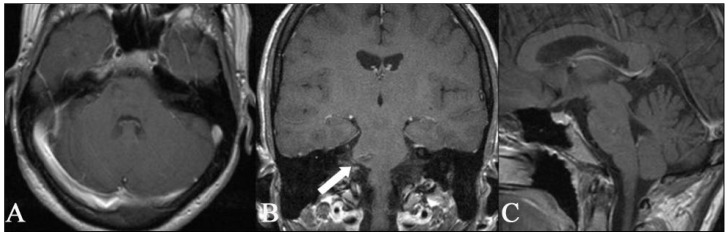
Postoperative axial (**A**), coronal (**B**), and sagittal (**C**) T1 images MR with contrast enhancement showing that complete removal of the superficial cavernous malformation of the pons was achieved. In the coronal image in the centre, the microsurgical entry site into the brainstem can be seen (white arrow). After surgery, no new neurologic deficits occurred in the immediate postoperative period or during the 3-month follow-up.

**Figure 9 medicina-59-01601-f009:**
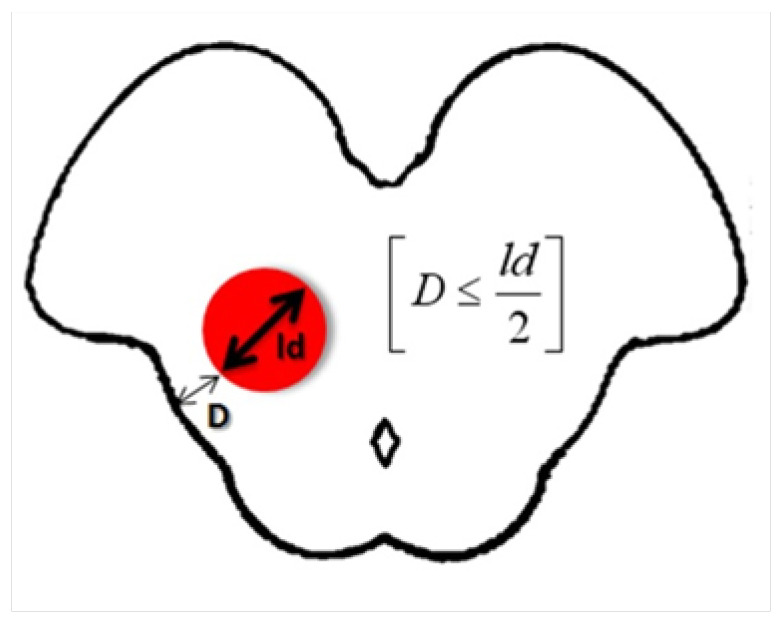
Illustration of the “Tuebingen brainstem cavernoma equation” used to determine whether a deep-seated lesion is close enough to the surface of the brainstem to be safely removed surgically. *D* is the shortest distance of the lesion (red circle) to the surface of the brainstem and should be equal to or less than half of the largest diameter (*ld*) of the lesion measured on an axial T1 scan.

**Table 1 medicina-59-01601-t001:** Distribution of surgical approaches according to the location of the cavernomas within the brainstem.

Location of Lesion	Approach	Craniotomy	Patient’s Position
Anterior Midbrain	Trans-Sylvian	Pterional	Supine
Lateral Midbrain	Supracerebellar infratentorial lateral	Retrosigmoid	Semi-Sitting
Medial Midbrain	Supracerebellar infratentorial medial	Medial suboccipital	Semi-Sitting
Anterolateral Pons	Cerebellopontine angle	Retrosigmoid	Semi-Sitting
Medial Pons	Transventricular (4th ventricle)	Medial suboccipital	Prone
Lateral Medulla	Trans-vellum medullare	Medial suboccipital	Prone
Medial Medulla	Subtonsilar Transventricular (4th ventricle)	Medial suboccipital	Prone

**Table 2 medicina-59-01601-t002:** Summary of pre- and postoperative clinical signs and neurological deficits of patients with deep-seated and superficial brain stem cavernomas (CN = cranial nerve) in the postoperative phase. *n* = number of patients.

	Symptoms (*n* = 34)	Headaches	Hypaesthesia	Gait Disturbance	Diplopia (CN VI)	Facial Palsy (CN VII)	Hearing Loss (CN VIII)	Difficulty Swallowing CN (IX–X)
Cavernoma Location								
Deep-seated BSCs preoperative		5.8% (*n* = 2)	14.7% (*n* = 5)	14.7% (*n* = 5)	14.7% (*n* = 5)	8.8% (*n* = 3)	0	8.8% (*n* = 3)
Deep-seated BSCs postoperative		0	11.7% (*n* = 4)	8.8% (*n* = 3)	11.7% (*n* = 4)	11.7% (*n* = 4)	0	11.7% (*n* = 4)
Superficial BSCs preoperative		17.6% (*n* = 6)	23.5% (*n* = 8)	23.5% (*n* = 8)	8.8% (*n* = 3)	11.7% (*n* = 4)	0	8.8% (*n* = 3)
Superficial BSCs postoperative		2.9% (*n* = 1)	14.7% (*n* = 5)	14.7% (*n* = 5)	14.7% (*n* = 5)	14.7% (*n* = 5)	2.9% (*n* = 1)	5.8% (*n* = 2)
New neurological deficit after surgery								
Deep-seated BSCs		0	0	2.9% (*n* = 1)	2.9% (*n* = 1)	2.9% (*n* = 1)		2.9% (*n* = 1)
Superficial BSCs				5.8% (*n* = 2)	5.8% (*n* = 2)	2.9% (*n* = 1)	2.9% (*n* = 1)	

**Table 3 medicina-59-01601-t003:** Modified Ranking Scale (MRS) of treated patients with superficial and deep-seated BSCs preoperative and in the long-term follow-up, with 5.8 years median follow-up for patients with deep-seated BSCs and 3.6 years median follow-up for patients with superficial BSCs. *n* = number of patients.

	MRS (*n* = 34)	MRS (Median)	MRS 0	MRS 1	MRS 2	MRS 3	MRS 4	MRS 5
Cavernoma Location								
deep-seated BSCs preoperative		1	0	8 (23.5%)	4 (11.7%)	0	1 (2.9%)	0
deep-seated BSCs postoperative		2	3 (8.8%)	7 (20.5%)	3 (8.8%)	0	0	0
superficial BSCs preoperative		2	1 (2.9%)	6 (17.6)	2 (5.8%)	2 (5.8%)	3 (8.8%)	0
superficial BSCs postoperative		3	2 (5.8%)	5 (14.7%)	3 (8.8%)	3 (8.8%)	1(2.9%)	0

## Data Availability

Not applicable.

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
