# Peer review of "The Brainstem Cavernoma Case Series: A Formula for Surgery and Surgical Technique"

_medicina, 2023, doi:10.3390/medicina59091601_

Round 1

Reviewer 1 Report

Dear Editor 

Thanks a lot for inviting me as the reviewer of the present manuscript entitled: "Deep Seated Brainstem Cavernomas: A Formula for Surgery and Surgical Technique" in which the authors have reported 34 cases of brainstem cavernous malformation which have been treated using neuronavigated microsurgery techniques. Authors have also introduced a new potential formula for decision-making in term of surgical removal of these cavernomas. I highly appreciate the effort put to this job as authors have addressed a really important topic in the field. However, there are some minor concerns about the results and their interpretation as well as the value given to the evidence which can be found in the following comments.

1- First, I think the title should be revised as this study has presented superficial cavernomas too. In addition, it is better to convey the term "a case series" as I explain it here after.

2- According to methodology, the study design should be mentioned in the method section to determine the level of evidence and I believe this is a case series study. Moreover, the setting(s) in which the study had been conducted as well as the time periods should be explained in detail in this section.
The diagram is a good idea; however, I think it could be revised a little. In the right side of diagram in terms of coincidental findings, when it comes to check the superficiality, if the answer is no, you have two choices: one to wait and watch or to check the progressivity. If there is no follow-up MRI, how can we directly check the progressivity. It needs to be clarified.

3- I am curious to see if authors can include the pathological assessment results of removed samples. That would help with the confirmed diagnosis.

4- Table 2 must be re-arranged in a way that makes it more understandable. Maybe exchanging rows and columns would be a good idea. I suggest the same for Table 3 and it should be moved and addressed in the result section, instead of discussion.

5-My most prominent concerns is about the so called "Tuebingen brainstem cavernoma equation". It seems that this formula has been sought retrospectively and authors have tried to find a mathematical way to prove their approach. I am not sure if it is an appropriate way for developing an equation as anybody may have his/her own point of view toward the lesion. For instance, I can consider distance from anterior or posterior surfaces and find a common way. I think this needs to be discussed within a interdisiciplinary group.
In addition, the low sample size is another drawback for generating the formula.

6- In line 280, authors have proposed surgery only if the risk of neural deficits are statistically lower than risk of hemorrhagic events. How can we measure these risks to calculate the statistical significance?

7- Results should be interpreted, concluded and generalized more cautiously according to observational level of evidence of this study and the low sample size.

Finally, I really appreciate again this opportunity and hope that my comments are not going to frustrate authors as I really want to help with more clear science transfer.

With best regards,
Reviewer

The manuscript would benefit from a slight English revision. It would be nice if a native English speaker can proof read it.

Author Response

Dear Sir or Madam,

Thank you for your review and your comments, which were very valuable for the improvements. We have revised the questions raised by the reviewers and have added the relevant text. The issues we addressed are indicated in red. Please find more detailed answers to your questions below. We hope that the revisions are now satisfactory for publication in Medicina.

Reviewer 1

COMMENT 1. Thanks a lot for inviting me as the reviewer of the present manuscript entitled: "Deep Seated Brainstem Cavernomas: A Formula for Surgery and Surgical Technique" in which the authors have reported 34 cases of brainstem cavernous malformation which have been treated using neuronavigated microsurgery techniques. Authors have also introduced a new potential formula for decision-making in term of surgical removal of these cavernomas. I highly appreciate the effort put to this job as authors have addressed a really important topic in the field. However, there are some minor concerns about the results and their interpretation as well as the value given to the evidence which can be found in the following comments.

First, I think the title should be revised as this study has presented superficial cavernomas too. In addition, it is better to convey the term "a case series" as I explain it here after.

ANSWER 1. Thank you for this comment. We have revised the title and added ‘a case series’. Since the article encompasses both superficial and deep cavarenomas, we have omitted ‘deep seated’ form the title. We propose the following new title:

The brainstem cavernoma case series: A formula for surgery and surgical technique.

We hope that the title is satisfactory, otherwise, we are happy to change it.

COMMENT 2. According to methodology, the study design should be mentioned in the method section to determine the level of evidence and I believe this is a case series study. Moreover, the setting(s) in which the study had been conducted as well as the time periods should be explained in detail in this section.

The diagram is a good idea; however, I think it could be revised a little. In the right side of diagram in terms of coincidental findings, when it comes to check the superficiality, if the answer is no, you have two choices: one to wait and watch or to check the progressivity. If there is no follow-up MRI, how can we directly check the progressivity. It needs to be clarified.

ANSWER 2. Thank you for the comment. We have supplemented this part.

The study was designed and conducted in the Department of Neurosurgery, College of Tuebingen, Germany. In this case series study, we collected data from a number of patients with deep-seated BS CMs and surgical outcomes after surgery. The operations were performed over a period of six years. Medical records and radiological images were reviewed. Surgical approaches, complications and treatment outcomes were analysed. All patients were fully informed and gave written informed consent after approval by the Ethical Institutional Review Board (Ethical Board number 3/maj/2022).

For the right part of the diagram, for progressivity check, MRI is essential. It is not possible to reach a decision without the MRI assessment. We have added this part to the diagram: In case of: progressive course of the disease (check with MRI follow-up once a year).

COMMENT 3. I am curious to see if authors can include the pathological assessment results of removed samples. That would help with the confirmed diagnosis.

ANSWER 3.

All the removed lesions were sent for pathohistological assessment, which is standard procedure when removing pathological tissue during brain surgery. In all cases, it was confirmed that the excised lesions were cavernomas.

COMMENT 4. Table 2 must be re-arranged in a way that makes it more understandable. Maybe exchanging rows and columns would be a good idea. I suggest the same for Table 3 and it should be moved and addressed in the result section instead of discussion.

ANSWER 4. The Table 2 was rearranged. We have also modified the Table 3 and we have moved it into the Results section.  In the Results section, we have established a new subheading: Neurological deficits after the operation. We hope that the tables are now more understandable, however, on our opinion, the former tables are better and if he respected reviewers agree, we would ask to keep the original form. Otherwise, we are happy to change them.

The modified Table 2:

Symptoms

        (n=34)

Cavernoma location

headaches

hypaesthesia

gait disturbance

diplopia

(CN VI)

facial palsy

(CN VII)

hearing loss (CN VIII)

difficulty swallowing

CN (IX – X)

deep-seated BSCs

pre-operative

5.8%

(n=2)

14.7%

(n=5)

14.7%

(n=5)

14.7%

(n=5)

8.8%

(n=3)

0

8.8%

(n=3)

deep-seated BSCs

post-operative

0

11.7%

(n=4)

8.8%

(n=3)

11.7%

(n=4)

11.7%

(n=4)

0

11.7%

(n=4)

superficial BSCs

pre-operative

17.6%

(n=6)

23.5%

(n=8)

23.5%

(n=8)

8.8%

(n=3)

11.7%

(n=4)

0

8.8%

(n=3)

superficial BSCs

post-operative

2.9%

(n=1)

14.7%

(n=5)

14.7%

(n=5)

14.7%

(n=5)

14.7%

(n=5)

2.9%

(n=1)

5.8%

(n=2)

New neurological deficit after surgery

Deep-seated BSCs

0

0

2.9%

(n=1)0

2.9%

(n=1)

2.9%

(n=1)

2.9%

(n=1)

Superficial BSCs

5.8%

(n=2)

5.8%

(n=2)

2.9%

(n=1)

2.9%

(n=1)

The modified Table 3:

    MRS (n=34)

Cavernoma

location

MRS (median)

MRS 0

MRS 1

MRS 2

MRS 3

MRS 4

MRS 5

deep-seated BSCs

pre-operative

1

0

8 (23.5%)

4 (11.7%)

0

1 (2.9%)

0

deep-seated BSCs

post-operative

2

3 (8.8%)

7 (20.5%)

3 (8.8%)

0

0

0

superficial BSCs

pre-operative

2

1 (2.9%)

6 (17.6)

2 (5.8%)

2 (5.8%)

3 (8.8%)

0

superficial BSCs

post-operative

3

2 (5.8%)

5 (14.7%)

3 (8.8%)

3 (8.8%)

1(2.9%)

0

COMMENT 5. My most prominent concerns is about the so called "Tuebingen brainstem cavernoma equation". It seems that this formula has been sought retrospectively and authors have tried to find a mathematical way to prove their approach. I am not sure if it is an appropriate way for developing an equation as anybody may have his/her own point of view toward the lesion. For instance, I can consider distance from anterior or posterior surfaces and find a common way. I think these needs to be discussed within a interdisiciplinary group. In addition, the low sample size is another drawback for generating the formula.

ANSWER 5. Thank you for your comment. We fully understand your concern but the formula is not an absolute formula and represents merely starting point for further evaluations in order to find a more objective, systematic and logical mathematical way to make a decision for surgery. It is supposed to be only the first step trying to make decisions for indications for surgeries of brainstem cavernomas easier. As we mention in our case series, further studies need to be performed in order to evaluate this formula. We have emphasised this more clearly in the manuscript.

COMMENT 6. In line 280, authors have proposed surgery only if the risk of neural deficits are statistically lower than risk of hemorrhagic events. How can we measure these risks to calculate the statistical significance?

ANSWER 5. Thank you or the question. We have supplemented the decision with the answer.

BS CMs are a unique subgroup of cavernous malformations. Although the statistical possibility for CM haemorrhage varies according to the literature, it is typical for the brainstem location to have higher haemorrhagic presentations, as well as technical challenges for their removal (1-4). Statistically, the average annual rate of CM haemorrhage is 0.7% to 1.1% per lesion in patients with no previous haemorrhage. This risk increases (to almost 4.5% and up to 60%) in patients who have had a previous lesion bleed (2). Approximately three to four years after a haemorrhagic event, the risk of further bleeding decreases. Of course, the risk of bleeding also depends on other factors, such as gender, the presence of a developmental venous anomaly and the location of the CM (1,2). As mentioned earlier, CMs located in the brainstem are more likely to bleed (3). According to the study by Kong et al, the 5-year cumulative risk of haemorrhage was 30.8% and was higher in subgroups when risk factors were present that helped predict potential bleeding in patients and were useful for treatment decision-making (4). According to known statistics, we recommend CM surgical removal in patients with risk factors that increase the possibility of bleeding. These include previous symptomatic bleeding, a positive family history, gender and age (4,5). All our operated patients had suffered at least one acute or previous bleeding episode. Therefore, according to the known literature, the risk of rebleeding in this group remains high and surgical treatment was recommended. This outweighed the potential risk of postoperative neurological impairment that may occur during surgery. In addition, all patients already had some neurological deficit due to the haemorrhage itself. Since the surgical mortality is reported to be 1.9%, the recurrence rate is more than 50-60%, which outweighs the risk even though surgery is performed in a problematic area such as the brainstem (10,23).

References

  1. Zyck S, Gould GC. Cavernous Venous Malformation. 2023 Mar 27. In: StatPearls [Internet]. Treasure Island (FL): StatPearls Publishing; 2023. PMID: 30252265.
  2. Atwal GS, Sarris CE, Spetzler RF. Brainstem and cerebellar cavernous malformations. Handb Clin Neurol. 2017;143:291-295.
  3. Li Z, Ma L, Quan K, Liu P, Shi Y, Liu Y, Zhu W. Rehemorrhage of brainstem cavernous malformations: a benchmark approach to individualized risk and severity assessment. J Neurosurg. 2022;139(1):94-105.
  4. Kong L, Ma XJ, Xu XY, Liu PP, Wu ZY, Zhang LW, Zhang JT, Wu Z, Wang L, Li D. Five-year symptomatic hemorrhage risk of untreated brainstem cavernous malformations in a prospective cohort. Neurosurg Rev. 2022;45(4):2961-2973.
  5. Geraldo AF, Alves CAPF, Luis A, Tortora D, Guimarães J, Abreu D, Reimão S, Pavanello M, de Marco P, Scala M, Capra V, Vaz R, Rossi A, Schwartz ES, Mankad K, Severino M. Natural history of familial cerebral cavernous malformation syndrome in children: a multicenter cohort study. Neuroradiology. 2023;65(2):401-414.

COMMENT 7. Results should be interpreted, concluded and generalized more cautiously according to observational level of evidence of this study and the low sample size.
Finally, I really appreciate again this opportunity and hope that my comments are not going to frustrate authors, as I really want to help with more clear science transfer.

ANSWER 7. Thank you for this comment. We have changed this in the text according to or best knowledge.

Reviewer 2 Report

I read with interest the study: "Deep Seated Brainstem Cavernomas: A Formula for Surgery and Surgical Technique" submitted by Guether C Feigl, Tomaz Velnar et al. for possible publication in Medicina (ISSN 1648-9144). 

The study is well conceived and well written, and surely deserves interest since it concern a difficult pathology in an extremely complex anatomical background. Brainstem surgery probably remains the most difficult region for a neurosurgeon. The idea of the formula is interesting, if even like every formula it should be viewed with caution and not as a rule.

I would suggest just some minor comments:

- Abstract: a bit too long, but well done. I would probably reduce the M&M section in this part, so it would become more fluid for readers.

- Introduction: nothing to concern. The purpose of the study is clear.

- Materials and methods: ok. The flowchart is essential in understanding the study. Images are well presented and clear. Figure 3 and 4 are very nice.

- Results: clear, ok.

Discussions: I would suggest to add some lines better discussing possible complications in this challenging anatomical region. Fequently some anatomical structures are "forgotten" and their clinical consequences underestimated, for example the triangle of Guillain and Mollaret, and the consequent hypertrophic olivary degeneration (you can find and interesting study by Tartaglione et al. DOI: 10.1007/s11547-014-0477-x). 

At the end of discussion section I would add some lines regarding limitations the current study, such as number of patients, etc.

Conclusion: maybe too long. I would suggest to cut the beginning: "The treatment of BS CMs remains a very complex matter, especially for the deep-seated 393 lesions. However, " and jump directly to the main findings of your study.

Author Response

Reviewer 2

I read with interest the study: "Deep Seated Brainstem Cavernomas: A Formula for Surgery and Surgical Technique" submitted by Guether C Feigl, Tomaz Velnar et al. for possible publication in Medicina (ISSN 1648-9144). The study is well conceived and well written, and surely deserves interest since it concern a difficult pathology in an extremely complex anatomical background. Brainstem surgery probably remains the most difficult region for a neurosurgeon. The idea of the formula is interesting, if even like every formula it should be viewed with caution and not as a rule.

I would suggest just some minor comments:

COMMENT 1. Abstract: a bit too long, but well done. I would probably reduce the M&M section in this part, so it would become more fluid for readers.

ANSWER 1. Thank you or the feedback and the review. We have shortened the Methods parts in the Abstract, as advised:

Abstract

Background and Objectives: Cavernous malformations (CM) are vascular malformations with low blood flow. Removal of brainstem CMs (BS) is associated with high surgical morbidity, and there is no general consensus on when to treat deep-seated BS CMs. The aim of this study is to compare the surgical outcomes of a series of deep-seated BS CMs with the surgical outcomes of a series of superficially located BS CMs operated on at the Department of Neurosurgery, College of Tuebingen, Germany. Materials and Methods: A retrospective evaluation was performed using patient charts, surgical video recordings, and outpatient examinations. Factors were identified in which surgical intervention was performed in cases of BS CMs. Preoperative radiological examinations included MRI with gadolinium contrast-enhanced T1 and T2 sequences and diffusion tensor imaging (DTI) to visualize brainstem fiber tracts. For deep-seated BS CMs, a voxel-based 3D neuronavigation system and electrophysiological mapping of the brainstem surface were used to customize the surgical approach and determine the shortest distance to the lesion and a safe entry point. All operated patients had suffered at least one acute or previous bleeding episode before surgical treatment was recommended. Results: A total of 34 consecutive patients with primary superficial (n=20 / 58.8%) and deep-seated (n=14 / 41.2%) brainstem cavernomas (BS CM) were enrolled in this comparative study. Complete removal was achieved in 31 patients (91.2%). Deep-seated BS CMs: The mean diameter was 14.7 mm (range, 8.3 to 27.7 mm). All but one of these lesions were completely removed. The median follow-up time was 5.8 years. Two patients (5.9%) developed new neurologic deficits after surgery. Superficial BS CMs: The median diameter was 14.9 mm (range, 7.2 to 27.3 mm). All but two of the superficial BS CMs could be completely removed. New permanent neurologic deficits were observed in two patients (5.9%) after surgery. The median follow-up time in this group was 3.6 years. Conclusion: The treatment of BS CMs remains complex. However, the results of this study demonstrate that with less invasive posterior fossa approaches, brainstem mapping, and neuronavigation combined with the use ofa blunt "spinal cord" dissection technique, deep-seated BS CMs can be completely removed in selected cases, with good functional outcomes comparable to those of superficial BS CM.

COMMENT 2. Introduction: nothing to concern. The purpose of the study is clear. Materials and methods: ok. The flowchart is essential in understanding the study. Images are well presented and clear. Figure 3 and 4 are very nice. Results: clear, ok.

ANSWER 2. Thank you for this comment.

COMMENT 3. Discussions: I would suggest adding some lines better discussing possible complications in this challenging anatomical region. Frequently some anatomical structures are "forgotten" and their clinical consequences underestimated, for example the triangle of Guillain and Mollaret, and the consequent hypertrophic olivary degeneration (you can find and interesting study by Tartaglione et al. DOI: 10.1007/s11547-014-0477-x). At the end of discussion section, I would add some lines regarding limitations the current study, such as number of patients, etc.

ANSWER 3. Thank you. In the Discussion, the complication section has been introduced and the limitations of our study were added. The recommended article was cited. We have added these new subheadings in the Discussion.

Possible complications after the BS CMs surgery

The brainstem is a difficult anatomical region and neurological sequelae are always possible due to both the lesion itself and the surgical procedure. The most common surgical complications are the standard complications that can occur with almost any neurosurgical procedure. In general, surgical complication rates vary between 2% and 9% (c). These include a mortality rate of 1.5%, a wound infection rate of 1.5% and a bleeding rate of 4.5% at the surgical site. Postoperative medical complications can occur in 3% to 9% of patients (c, b). Complications related to the cranial nerves, sensory and motor pathways, cerebellar mutism and cerebrospinal fluid circulation are possible with brainstem surgery (b). In our patients, the complications we noted resulted in headache, hypaesthesia, gait disturbances, cranial nerve deficits, incomplete removal of the CM, hydrocephalus and degenerative motor fibre disease of unknown cause.

An important and rarely reported complication is olive degeneration (1,2). It is a relatively common consequence of posterior fossa surgery, especially in children treated for high-grade tumours (3-5). Olive degeneration is caused by lesions of the posterior fossa or brainstem involving the dentato-rubro-olivaria pathway, also known as the Guillain-Mollaret triangle. Clinical and radiological features of this disorder include ocular myoclonus, gaument tremor, nystagmus, ataxia and vertigo. MRI shows T2 hyperintensity of the inferior olive complex on magnetic resonance imaging. Symptomatic degeneration of the olives can complicate recovery in patients with posterior fossa or brainstem lesions. It most commonly occurs after posterior fossa surgery, although other cases are also important, such as posterior fossa tumours, traumatic brain injury and cerebellar infarction. The most common reported cause is posterior fossa surgery secondary to cavernomas. Although most patients may remain asymptomatic, degeneration of the orbit complicates postoperative recovery (2-5). In our patient group, we did not directly evaluate olive degeneration clinically. However, complications in our group included gait disturbance and diplopia, so some of the cases might be related to some kind of olive degeneration. Because disruption of the dentato-rubro-olivary tract can complicate patient recovery, neurosurgeons should be aware of this complication to avoid misdiagnosis and unnecessary investigations.

References  

  1. Gradišnik L, Bošnjak R, Bunc G, Ravnik J, Maver T, Velnar T. Neurosurgical Approaches to Brain Tissue Harvesting for the Establishment of Cell Cultures in Neural Experimental Cell Models. Materials (Basel). 2021 Nov 13;14(22):6857.
  2. Anetsberger S, Mellal A, Garvayo M, Diezi M, Perez MH, Beck Popovic M, Renella R, Cossu G, Daniel RT, Starnoni D, Messerer M. Predictive Factors for the Occurrence of Perioperative Complications in Pediatric Posterior Fossa Tumors. World Neurosurg. 2023 Apr;172:e508-e516.
  3. Tartaglione T, Izzo G, Alexandre A, Botto A, Di Lella GM, Gaudino S, Caldarelli M, Colosimo C. MRI findings of olivary degeneration after surgery for posterior fossa tumours in children: incidence, time course and correlation with tumour grading. Radiol Med. 2015;120(5):474-82.
  4. Onen MR, Moore K, Cikla U, Ucer M, Schmidt B, Field AS, Baskaya MK. Hypertrophic Olivary Degeneration: Neurosurgical Perspective and Literature Review. World Neurosurg. 2018 Apr;112:e763-e771.
  5. 5. Ogut E, Armagan K, Tufekci D. The Guillain-Mollaret triangle: a key player in motor coordination and control with implications for neurological disorders. Neurosurg Rev. 2023 Jul 20;46(1):181.

The limitations of our study

Of course, the study also has some limitations. The first limitation is the relatively small number of patients. However, we can say that CMs in the brainstem are not a common pathology compared to CMs in supratentorial areas and cerebellum. The patients with BS CMs represent a specialised subgroup that tends to settle in specialised and dedicated neurosurgical centres that frequently treat such pathologies, as was the case in our case. The solution here would be to expand the retrospective assessment framework and thus capture more patients. Secondly, although our patients were diagnostically well prepared for surgery, DTI, which is of great importance in brainstem CMs, was performed only in patients operated on after January 2008. This technique was unfortunately not available in our clinical department before. We believe that safety would increase if diagnostic imaging were even better. Third, as mentioned above, the Tuebingen brainstem cavernoma equation helps in deciding when deep-seated BS CMs should be operated on. However, it must be interpreted and applied with caution and is not a simple formula for choosing surgery. Each patient must be interpreted individually, taking into account all risk factors, including family history, risk of rebleeding, location of CM and consideration of possible neurological sequelae that may result from surgery. Fourth, we can focus more on the new complications following surgery in such challenging anatomical areas, such as the olive degeneration described earlier.

COMMENT 4. Conclusion: maybe too long. I would suggest cutting the beginning: "The treatment of BS CMs remains a very complex matter, especially for the deep-seated 393 lesions. However, " and jump directly to the main findings of your study.

ANSWER 4. Thank you. This part has been changed in the article:

Conclusions

The treatment of BS CMs remains a very complex matter, especially for the deep-seated lesions. However, our series shows that microsurgical resection of deep-seated BS CMs results in good functional outcomes. The results of this study show that by using the "spinal cord dissection technique" in combination with minimally invasive posterior fossa approaches, brainstem mapping, and neuronavigation in the hands of an experienced team, both superficial and deep-seated BS CMs can be completely removed in the majority of cases with a good functional outcome. The current data do not indicate that there are alternative treatments for these brainstem lesions that provide similarly good results.
